# The Implementation of National Dementia Plans: A Multiple-Case Study on Denmark, Germany, and Italy

**DOI:** 10.3390/ijerph181910220

**Published:** 2021-09-28

**Authors:** Nadia Céline Boeree, Claudia Zoller, Robbert Huijsman

**Affiliations:** 1Erasmus School of Health Policy & Management, Erasmus University Rotterdam, P.O. Box 1738, 3000 DR Rotterdam, The Netherlands; nadia_boeree@hotmail.com; 2Center for Social & Health Innovation, Management Center Innsbruck, Universitätsstraße 15, 6020 Innsbruck, Austria; claudia.zoller@mci.edu

**Keywords:** Global Action Plan, national dementia plan, policy implementation, public policy, qualitative research

## Abstract

In a theory-driven, qualitative, multiple-case study, we examined the implementation of national dementia plans (NDPs) in Denmark, Germany, and Italy to determine how stakeholders involved in implementing the NDPs evaluate contextual factors in light of the World Health Organization’s seven action areas of dementia-focused policy. To analyse the NDPs, we used a driver diagram of large-scale change and conducted both document analysis and semi-structured interviews, after which we performed three-way open coding to analyse the methods. The results show that the implementation of NDPs has increased awareness of dementia in all three countries by positioning the disease as a national public health concern. The study also reveals that resources, the use of change theory, and the monitoring of implementation and fragmentation are crucial factors to consider for countries when implementing NDPs. Although stakeholders find the NDPs useful, many challenges remain for their successful implementation due to the highly limited means for implementation and evaluation. Moreover, present NDPs all lack a communication plan that encompasses all layers of society to address ways of achieving change. Patients with dementia, and their informal caregivers, should be included more in the design, implementation, and evaluation of NDPs in order to improve their usefulness and effectiveness.

## 1. Introduction

As populations around the world continue ageing, the number of people living with noncommunicable diseases, such as dementia, has risen [1]. Worldwide, approximately 50 million people suffer from dementia, a figure that is expected to increase to 82 million by 2030, and 152 million by 2050 [1]. *Dementia* is a general term for a multitude of brain diseases that are mostly progressive in nature and affect memory, behaviour, and other cognitive abilities. Representing 60% to 70% of all cases, the most common form of dementia is Alzheimer’s disease, followed by vascular dementia, dementia with Lewy bodies, and frontotemporal dementia [2].

Because dementia significantly interferes with a person’s ability to maintain the activities of daily life, the disease is a chief cause of disability and dependency amongst patients. Dementia can also impact informal caregivers, including spouses, adult children, children-in-law, and/or friends, and poses a range of physical, psychological, social, and economic consequences for them and society at large [3]. Many such consequences are exacerbated by a lack of awareness and understanding of dementia, which can intensify the stigma of the disease and its patients, as well as raise hurdles for its diagnosis and care [1]. Moreover, as the number of people with dementia rises, so do the costs of care and support.

Although dementia has long been a neglected topic, its status has changed as it has become viewed as not only a national, but also a global, public health priority [4,5]. To meet the needs of people with dementia and their caregivers, which include adult children, spouses, daughters, sons, and friends, governments the world over face pressure to incorporate dementia into public policy [5], as well as to develop and implement national dementia plans (NDPs) able to sustain high-quality care and support for people with dementia, and ensure that health and social care systems for them are sufficiently structured and funded [6]. In Europe in particular, increased awareness of the need to address dementia in public policy has urged numerous countries to implement NDPs [7], and Alzheimer Europe has contributed to the knowledge of dementia with updated reports on national dementia policies in the Dementia Monitor 2020 [8,9]. In parallel, research has been increasingly conducted on the importance of NDPs and the examination of dementia-related topics, such as translational research and clinical research [5,6], while awareness of the need to join forces in research and data sharing addressing dementia has increased as well [10].

On the global scale, a report from the World Health Organization (WHO) titled “A Global Action Plan on the Public Health Response to Dementia (GAPD)” has encouraged all countries to develop and implement national dementia policies by 2025 [2]. Although the GAPD is founded upon theories and concepts, practical steps are required for the most effective implementation of NDPs. Thus, the GAPD describes seven action areas for establishing and implementing NDPs in a structural way, namely, “dementia as a public health priority”, “dementia awareness and friendliness”, “dementia risk reduction”, “dementia diagnosis, treatment, care, and support”, “support for dementia carers”, “information systems for dementia”, and “dementia research and innovation” [2]. Even though the WHO has also outlined specific actions that need to be taken in response to dementia, those actions require the active participation of all stakeholders involved, including national and regional partnering agencies, research institutes, and associations of patients and suppliers. Added to that, the WHO specifies that countries need to assess their capabilities and challenges in defining the roles and actions required by the stakeholder groups involved in addressing dementia within their borders.

To that purpose, the structural framework provided by the WHO presents targets for global dementia-oriented achievements [2], largely as a guide for each country to set its own national goals [11]. Using the framework, each country can assess how the targets should be met and modified depending on its national circumstances. At the same time, the WHO recognises that each country faces challenges when implementing actions in each action area, including differences in patterns of dementia, the roles of government actors, and/or care and support networks [11]. Despite all of those differences and factors, the WHO has adopted an overarching goal “to improve the lives of people with dementia, and their informal caregivers, while decreasing the impact of dementia on them as well as on communities and countries” [2].

Nevertheless, it remains unclear whether countries can manage those differences themselves and whether the critical care pathway outlined by the WHO suits each different country. Cahill [11] has also indicated that the challenges that countries face when implementing the WHO’s seven action areas deserve more attention in research. In our study, we did exactly that, specifically from a governmental perspective focused on all relevant internal and external factors, all actors involved in the NDP, and all stakeholders interested in the implementation of a particular country’s NDP. After all, the picture of internal and external factors remains incomplete and can easily differ between countries and stakeholder groups. Whereas *internal factors* are factors affecting implementation by the government (e.g., the initiative itself or the policymakers involved), *external factors* are factors affecting the context within which the initiative is implemented, including external organisers (e.g., the organisations and/or clinics involved).

Against that background, the objective of our qualitative study was to contribute to the understanding of all factors relevant to implementing NDPs. Our study was guided by the following research question: How do the stakeholders involved in the implementation of NDPs evaluate contextual factors in their country in light of the WHO’s seven action areas for dementia policy?

### Theoretical Framework

Although numerous studies in implementation science have identified internal and external factors that influence the implementation of policy, none have addressed factors related to implementing NDPs [11]. For starters, Nilsen’s [12] classification of the different implementation theories, models, and frameworks present in academic research distinguishes five core categories of implementation research: (1) process models that describe and/or guide the translation from theory to practice; (2) determinant frameworks about the factors that support or hinder implementation; (3) classical theories about aspects of implementation originating from psychology, sociology, and organisational theory; (4) implementation theories from implementation researchers; and (5) evaluation frameworks to determine the success of implementation. Because the goal of our study was to understand which factors can influence the implementation of NDPs and, therefore, their outcomes, we applied the determinant framework to comprehend and clarify the outcomes of the implementation of various NDPs [12].

As the mentioned classification scheme suggests, several frameworks and models are available for reviewing and shaping factors of policy-related processes [13,14]. However, few frameworks discuss all the factors that could improve initiatives for large-scale change [15]. Nevertheless, that overlooked focus is crucial for studies such as ours, because implementing an NDP aims to improve the experiences and outcomes for people with dementia on the national scale. It is therefore imperative that the framework chosen for such research provides a comprehensive overview of factors that influence the implementation of especially large-scale improvement initiatives in health systems [16]. Within determinant frameworks, context is also significant. In our case, the context refers to the circumstances or settings in which the phenomenon of an NDP takes shape, even if accounting for the context makes identifying factors and their complexity more challenging.

For our study, we adopted the comprehensive framework developed by Perla et al. [17], which addresses several factors that can affect the implementation of policies, provides a context for understanding those factors, and indicates ways of strengthening them. Derived from the literature, the framework synthesises various models, theories, and frameworks into a driver diagram consisting of four primary drivers and 15 secondary drivers that can altogether be viewed as a checklist for implementation [17]. The primary drivers are: (1) characteristics of an NDP concerning planning and infrastructure; (2) individual, group, organisational, and systemic factors, as well as contextual factors borrowed from Ferlie and Shortell [18]; (3) the underlying theory and process of change for involving all partners; and (4) performance measures and evaluation [17]. Within those drivers, because certain factors have less significance when observing an NDP as a large-scale initiative in healthcare, some factors have been added or removed from the framework. In Figure 1, the factors are divided into primary (in bold) and secondary factors that affect the implementation of NDPs.

## 2. Materials and Methods

### 2.1. Design

Our multiple-case study was designed to compare how different countries have implemented NDPs, as well as to gain insight into variations in their implementation processes, all by using qualitative data to determine how the NDP initiatives have been implemented and why their implementation has differed. By using a constant systematic way of investigating all of the different cases, generalisability was attainable; thus, although the sample was not representative of all countries, the study and its findings are conceptually generalisable. In that way, the outcomes of our study stand to illuminate contexts of healthcare that differentiate the original cases that we examined, even when countries that have used the same methods have experienced similar outcomes. Last, we also employed multi-method triangulation to minimise biases and flaws.

### 2.2. Case Selection

For our study, we identified three countries that would afford a sound representation of the different healthcare structures and cultures in Europe. Although we did not consider the east–west gradient because of the lack of (English-language documentation concerning) NDPs in eastern European countries, the north–south gradient that we considered is often taken into account in comparative research in order to include variations in levels of education, levels of income, and health status or access to healthcare [19]. Along those lines, we selected Denmark as a northern representative, Italy as a southern representative, and Germany as a representative in between. Although Denmark has historically had a decentralised health system, reforms in 2007 centralised healthcare decision-making in the country [20]. In Germany, however, the healthcare system became decentralised in 2009 [21]. In Italy, the national government has created general objectives for the healthcare system, although the regional governments are ultimately accountable for its delivery [22]. On the whole, because the three countries are at different stages of implementing their NDPs and represent a mix of centralised and decentralised strategies in Europe, they were deemed to be suitable as case studies.

### 2.3. Data Collection

#### 2.3.1. Document Analysis

Document analysis was performed on studies published on the PubMed and Google Scholar databases. To find significant results concerning the three countries’ NDPs, terms such as “national dementia strategy”, “national dementia plan”, “dementia policy”, and “dementia interventions” were used (see Table 1, including the referral codes). The respondents that we recruited for interviews via snowball sampling were also asked to supply relevant national documents for analysis. The data collected from document analysis were used to compare the three countries’ NDPs with the GAPD, gain knowledge about the implementation of the NDPs in the countries, and determine possible factors to prepare for the semi-structured interviews.

#### 2.3.2. Semi-Structured Interviews

The interviews were semi-structured around a set of prearranged topics that followed from Perla et al.’s framework [17] (see Appendix A). Even so, additional topics could evolve during the interviews, which provided the necessary space for the interviewees to describe their experiences. The respondents were interviewed via an online communication platform (i.e., Zoom or Microsoft Teams) due to geographical distance and restrictions imposed for the COVID-19 pandemic. Prior to interviews, each interviewee received an email containing information about the research objectives and an informed consent form seeking their permission to transcribe and cite the interview [34].

#### 2.3.3. Respondents

The study population for the semi-structured interviews consisted of one respondent from each key stakeholder group involved in implementing the NDP in each of the three countries (see Table 2) as per WHO [2] recommendations. The sample thus included representatives of national agencies (e.g., ministries and working groups) and associations of care providers and patients (e.g., dementia federations or associations). Although the WHO has also recommended including national research institutes as stakeholder groups, such representatives in all countries were either unwilling or unable to participate.

### 2.4. Data Analysis

After the interviews, the audio-recordings of the interviews were transcribed. To enhance the data’s accuracy, the respondents were asked to check the transcript as a form of member check [35]. The transcripts were analysed using a three-step coding process involving open, axial, and selective coding [36], in that order, all using the qualitative data analysis programme ATLAS.ti (version 9). In open coding, the transcripts were read, and various keywords and codes were linked to the text fragments. Next, the keywords and codes were combined into labels during axial coding. By bundling all of the labels into main concepts, a code tree was created (see Appendix A). Third, selective coding involved analysing the linkages between the major concepts. The documents representing the three countries were coded in the same way. Ultimately, only the most relevant codes were considered to derive important quotations that illustrate the results.

## 3. Results

### 3.1. Three NDPs

Denmark’s healthcare system operates at three administrative levels. First, the national government is responsible for establishing a framework for healthcare services, planning, monitoring, licensing, and the allocation of the budget depending on the demography of the region or municipality. The delivery of services, management, and the financing of hospitals and other service providers, by contrast, is the responsibility of the five regions governed by democratically elected councils. Last, the municipalities are responsible for finances and the provision of primary healthcare services and elderly care, including in nursing homes [37,38]. Denmark’s first NDP was implemented in 2010 and continued until 2014, and the country is currently following its second NDP, an enhancement of the first plan. Published in 2017, Denmark’s current strategy is community-based, focuses on dementia-friendly municipalities, and will continue until 2025 (DD1). The process of developing the NDP was comprehensive and inclusive; actors, such as associations, research institutes, ministries, and regions, represented different layers of society. Although developing the NDP was initially a professional process, it gradually transformed into a political one as part of the government’s phase-based plan (DD3). In the end, “The Ministry of Health was the contributor—the one who was writing the plan and sort of the owner of the process” (A1). For most of the initiatives, the National Board of Health has had the ultimate responsibility and could be co-chaired with other organisations or associations (DD1, A1, A3).

Germany’s healthcare system consists of different actors who have decision-making power, including the federal and state governments, along with self-regulating organisations of payers and providers [39]. Although the federal government has broad regulatory power, its power is less in the delivery of healthcare. The Federal Joint Committee, a decision-making body supervised by the Federal Ministry of Health, determines which services need to be covered and is responsible for measuring quality for providers and regulating ambulatory care capacity [40]. The Committee counts 13 members with voting rights, “consisting of five representatives from associations of sickness funds, five from associations of providers, and three unaffiliated members” [40]. Additionally, five patient representatives without the right to vote are included as advisory members. Thus, the federal government was ultimately responsible for creating the NDP.

Germany ranks amongst the most recent countries to implement an NDP, which it accomplished in 2020. Its NDP is a system of local networks that involves coordination between medical and social care structures and cooperation with different partners. The strategy was built on the “Alliance for People with Dementia”, an initiative that ran between 2014 and 2018 and implemented its agenda, titled “Together for People with Dementia” (DG1). The strategy was further anchored in the coalition agreement of the current federal government and adopted by the federal cabinet (DG1). In creating its NDP, Germany relied heavily on a network approach, the need for an NDP, and the collaboration of multiple actors representing different layers of society. Its implementation followed a similar approach, with the German Alzheimer Society serving as co-chair and actively supporting the process (DG1). Added to that, “For each measure, different stakeholders were identified who are responsible for the implementation. By some measures, one stakeholder is identified to coordinate the actors for the measures” (B3). By giving stakeholders that responsibility, internal control over the NDP’s implementation increased.

Italy’s healthcare system is organised on a quasi-federal state level, with every region and local authority having the obligation to organise, deliver, and provide healthcare services. In total, Italy has 19 regions and two autonomous provinces, each responsible for its own delivery of care. Together with the national government, regions are responsible for the quality of care and for ensuring that the healthcare services are provided within a certain timeframe. Although global strategies are created at the national level, their implementation has to be approved by a joint state–regional standing committee. Ultimately, the regions enjoy significant freedom in establishing the macrostructure of their health systems [41].

Likewise, the Italian NDP also required approval by a joint committee of state and regional actors. Before the plan could be approved, long-lasting negotiations between the actors representing different levels and sectors involved addressed ways of ensuring the NDP’s financial viability and the need to maintain coherence despite differences within regions and local communities. Regardless of the complexity of involving both national and regional actors, Italy’s NDP was drafted and approved in 2014 [42]. Although most of the writing was performed by the national Ministry of Health, the Ministry worked closely with all stakeholders involved (DI1) and, during implementation, the stakeholders involved depended on objective and corresponding guidelines. Whereas some parts of the plan needed to be implemented at the national level, including with the support of the National Dementia Observatory (DI5), the NDP’s core parts required a regional plan, including diagnostic therapeutic care pathways for people with dementia [43].

### 3.2. Seven Action Areas from the GAPD

A close comparison of the three NDPs with the GAPD (see Table 3) reveals that all comply with Action Area 1 by viewing dementia as a public health priority, as demonstrated by their implementation of the NDPs. All three countries have also addressed Areas 2 (i.e., awareness and friendliness), 4 (i.e., diagnosis, treatment, care, and support), 5 (i.e., support carers), and 7 (i.e., research). For instance, Germany’s NDP prioritises “developing and establishing dementia-inclusive communities to enable people with dementia to participate in society, supporting people with dementia and their relatives, advancing health and long-term care services for people with dementia, and promoting excellent research on dementia” (DG1). Those priorities have been subdivided into 27 objectives with approximately 160 specific actions (DG1).

Italy’s NDP consists of four principal objectives, all of which overlap with Action Areas 2 and 5 in the GAPD—to “promote health and social care interventions and policies, create and strengthen the integrated network of services for dementia based on an integrated approach, implement strategies for promoting appropriateness and quality of care, and improve the quality of life of persons with dementia and their families by supporting empowerment and stigma reduction” (DI1). Those objectives have been outlined into 17 different actions (DI1). Compared with the other two countries, Italy’s NDP is decentralised, because the regions have autonomy and can adjust according to their regional needs. Additionally, Italy has approved three documents that specify particular objectives, including the National Dementia Observatory.

Action Area 2 (i.e., awareness and friendliness) is embedded in Germany’s NDP. Although Italy and Denmark have included dementia-friendly initiatives, they lack a national dementia awareness-raising campaign. Nevertheless, Denmark allocated a budget for local “information campaigns” (DD1) holding that “all 98 municipalities in Denmark should be dementia friendly, more people with dementia need to be identified, with 80% needing to have a specific diagnosis, and care and treatment need to be improved to reduce the consumption of antipsychotic medicines amongst people with dementia by 50% before 2025” (DD1).

By contrast, Action Area 3 (i.e., risk reduction) is not addressed in any of the plans; however, the three countries do address the topic in separate national plans—for example, in Italy’s “National Prevention Plan 2014–2018” [42].

Action Area 6 (i.e., information systems) has been implemented differently in all three NDPs. Italy does not address clear indicators to monitor the plan but has a monitoring table that meets as necessary and makes new guidelines accordingly (DI6). Denmark has clear indicators for certain areas, mostly ones in which quantitative data are easily available (DD1; DD2). Germany has addressed Area 6 in its NDP; however, the actual indicators have been stated in a later report (DG4), which states that a questionnaire will be sent to all actors involved to measure the plan’s progress and areas for potential improvement (DG5).

### 3.3. Stakeholders’ Perspectives on NDPs

Overall, the respondents agreed that implementing NDPs is useful because having such plans is important and because NDPs raise awareness of dementia as a whole, chiefly by situating it as a public health priority in the respective countries. On that topic, typical statements from the interviews include “awareness of dementia is rising” (A2), and “The NDP and the measures that are formulated have the potential to be extremely useful” (B4).

The actual results and effectiveness of the NDPs, however, can be disputed depending on the perspective, the country, and the stakeholder group. The respondents from the patients’ associations stated that the results and effectiveness of the NDPs have to be evaluated by the patients, although forming a representative sample of patients remains difficult. Not only do patients have different perspectives, but the healthcare system differs from region to region:
Nobody has collected a representative perspective to say whether it was successful or not. But of course, we get the feeling from our experience, from the feedback of the people who call us, and from our member associations at the local level that its success has depended on the region and each beneficiary’s personal experience (B3).

Although respondents from Germany described the NDP as useful and effective, they also recognised that improvements are possible. Because Germany’s NDP has only been in place for half a year, it has been difficult to gauge the results. What respondents emphasised can be learned from the German strategy is the high-level of collaboration. In particular, a “Network National Dementia Strategy” was established for regular meetings, with all stakeholders involved, to invite practical examples and further discuss certain issues.

Quite a lot of actors joined forces to develop a strategy for Germany. And that was a pivotal step, that we can build on in the future as well. Strategies such as the “Network National Dementia Strategy” are also quite useful, because they can help to facilitate exchange on certain topics and might reveal good solutions for certain issues in the future as well. Exchange is always very good (B1).

Meanwhile, Denmark’s significant progress in positioning dementia as a public health issue within its government has bolstered awareness of dementia:
We saw a massive spike in diagnosed cases in 2018–2019. When we speak to our co-workers in memory clinics at hospitals, there was a rush of people sent there to get diagnosed, which is one of the goals of the plan. That part of the plan really paid off (A2).

However, getting results from other initiatives, whether effective or not, can help to further attain the objectives and goals: “There have been a lot of useful initiatives with a lot of good results. More importantly, we learned a lot as well on the way” (A1).

In Italy, the NDP has been regarded as useful because it placed dementia on the agenda of policymakers by making it a public health priority. Nevertheless, its priority has varied from region to region, some of which have more autonomy than others. Having an NDP has also sharpened the focus on dementia as a public health issue in all regions: “If there weren’t an NDP, then other regions would not have started to recognise the issue of dementia. With the NDP, all regions were pushed to do something because it’s a national law, and no one can be excused from national law” (C4). However, without funding, the steps of practical implementation have been difficult to achieve, even if Italy was one of the first countries with an NDP:
The effects are very limited now, because without funding, an implementation system, and a monitoring system, the usefulness of the plan remains very limited, of course. So, it’s sad to say, but the plan at the moment, especially from a cultural standpoint the practical effect of the plan has remained limited (C2).

### 3.4. Factors Influencing the Implementation of NDPs

Although the NDPs have been organised differently in the three countries, all have sought to address the phenomenon of dementia from a holistic approach—that is, by targeting not only people with dementia but all actors in society affected by the disease. A typical statement from respondents was “An NDP implies the participation of most of the stakeholders involved” (C2). The focus of the NDPs is, thus, not solely the disease but the complete physical and mental well-being of people with dementia, as well as society, because “in nearly all fields in which health is concerned, we need a partnership with the social dimension and in dementia more than in other situations” (C1).

The NDPs have not been supported by strong management plans but have had a clear leader and/or responsible stakeholders to oversee the process. Overall, the respondents addressed objectives handled by the countries’ health ministries in greater detail than objectives in the decentralised hands of the regions or local organisations. Although information from Denmark did not include how the initiatives have been managed or what each stakeholder has needed to implement, some regions clearly have a regional plan, which has made the initiatives clearer. Denmark’s National Board of Health, for example, has reserved a specific part of its budget for hospitals and councils to apply for as a means to increase local ownership. Those organisations could receive funding provided that they have a firm proposal for working in the field of dementia with increased knowledge and competencies. By comparison, Germany has increased internal control by managing the initiatives at the federal level and giving all stakeholders tasked with an objective the responsibility of implementing the measure: “Full responsibility for achieving the objectives or measures lies with the stakeholders. Often several stakeholders work together to integrate their perspectives” (B2).

Because NDPs are arranged at the national scale, many respondents have worked primarily at the national level, whereas fewer stakeholders, groups, and actors were involved at the micro level. However, stakeholders from all levels of society were reported to be involved throughout the entire process of creating and implementing the NDPs: “Something that I will suggest to everybody is that the stakeholders are part of the problem but also part of the solution” (C1). As such, NDPs have to accommodate the diversity of the subsystems involved and, in turn, their varying capabilities. In general, however, the respondents agreed that local administrators and local organisations lack the power to implement the objectives of the NDPs adequately: “Many local councils have found that they didn’t really have enough staff or implementation power to succeed with all of the projects” (A1).

To get more stakeholders involved, and to expand knowledge of dementia, local champions and change agents have been used as well. Germany uses experts for its NDP, Italy uses researchers and employees of the Ministry of Health and the Institute of Health, and Denmark maintains NGOs, researchers, associations, and employees from the government. Although the NDPs have thus been driven by different stakeholders in each country, none of the countries has a team that coordinates the specific strategy. Moreover, they all lack a communication plan that encompasses all layers of society to address ways of achieving change. Although Denmark has not had such a plan, A1 reported that “the minister’s office had a communication plan” that involved all stakeholders. By contrast, Germany had an unofficial communication plan, with a strategy consisting mostly of negotiations with all actors involved in each measure to encourage them to use the channels at their disposal to raise awareness about the NDP. Finally, in Italy, the lack of a clear communication plan, and the minimal involvement of all-but-essential stakeholders, was characterised as a flaw in its NDP: “That’s the kind of thing lacking in Italy. I’d say there’s no national communication plan. There’s no direct communication” (C2).

In Italy, as well as Denmark, there has been no national dementia awareness campaign, either. In Italy, the absence can be explained by a lack of financing from the federal government. Although Italy’s NDP was approved in 2014, the first round of funding may not even be issued in 2021: “It’s clearly stated that part of the money that will finance the NDP is for our health minister to run a campaign for dementia awareness” (C3).

From the beginning, the NDPs in all three countries have also lacked performance measures and evaluations of their initiatives. However, all stakeholders emphasised the importance of a measurement and feedback system for the implementation of NDPs. In Germany’s NDP, which initially had no infrastructure in place to collect data, stakeholders recognised its importance, and following persistent pressure, a questionnaire was created. Because Germany’s NDP remains in its infancy—it was launched in summer 2020—no data regarding its usefulness are available. Denmark has no monitoring plan for the NDP overall but monitors and evaluates all 23 different initiatives separately. Italy also lacks a national monitoring plan or system, possibly because of the lack of funding for national data infrastructure from the national government. However, it recognises potential indicators that are of value for future monitoring. In Italy, implementing NDP guidelines and collecting data thus fall to the regions. In response to such shortcomings, most respondents advocated for a European or international research strategy or monitoring plan, which they expected would enable the countries to learn from each other as to whether objectives can be achieved, and how to implement an NDP in the best ways possible.

## 4. Discussion

To the best of our knowledge, our study represents the first in-depth research into factors that could contribute to the implementation of NDPs according to different stakeholders. Document analysis revealed that not all aspects recognised by the WHO are included in the NDPs of Denmark, Germany, and Italy. In addition, the interviews revealed that although stakeholders find the NDPs useful, many challenges remain for their successful implementation because of the highly limited means for evaluating the results.

The implementation of NDPs has increased the awareness of dementia in all three countries by positioning the disease as a national public health concern. As a consequence, Denmark recorded a significant spike in diagnosed cases, which ranked amongst its NDP’s major goals. However, the fragmentation of implementation and the diffusion of responsibilities between different regional and national entities have increased the difficulty of successful implementation in all the countries. Because responsibilities are mostly deferred to the regions without a clear overall management strategy, regional actors often lack the power to take the necessary actions while navigating complex subsystems, which has reduced the transparency of the objectives and the overall process. Differences in Denmark were also observed in stakeholder involvement and funding for NDPs and, in the latter case, the lack of funding owing to changes in political power or tedious bureaucratic processes add to delays and can even hinder implementation. Beyond that, awareness of the need to include all stakeholders in implementing NDPs has sometimes been absent among decision makers. While all three countries rely on stakeholders with varying backgrounds and degrees of involvement, none have communication or coordination plans to ensure the involvement of all responsible stakeholders and layers of society.

Above all, the lack of standardised evaluation methods with limited means of data comparability restricts the informative value of NDPs and their success. Given the absence of empirical evidence, no clear performance evaluations or conclusions can be made regarding the effectiveness or quality of the various factors of NDPs or their overall objectives. Although minor efforts have been made, the lack of funding imposes significant restrictions.

Implicitly, the factors addressed by the stakeholders aligned with the themes from the conceptual framework, including resources, the use of change theory, monitoring, and fragmentation. To enhance the real-world results and usefulness of the NDPs, it could be valuable to implement the NDPs more in line with the GAPD’s seven action areas [2]. Furthermore, the move towards taking a network approach to implement the NDPs seems to be able to tackle many of the challenges plaguing today’s healthcare systems.

Although the countries that we selected to examine represent the different healthcare systems in Europe, and the different implementation phases of NDP policy, the limitations of our selection warrant attention. First, because the NDPs show great variation in structure and content, it remains difficult to conclude whether the findings are fully generalizable, for the research was conducted in only three countries along the north–south gradient of Europe. Second, all three NDPs focus on clinically diagnosed dementia and, therefore, lack strategies for identifying prodromal stages of dementia and the inclusion of biomarkers. Third, the sample size for the interviews was limited because of time constraints and multiple rejections from prospective respondents in light of COVID-19-related priorities at their organisations. The respondents represented different organisations that work with, or have an interest in, their respective NDP, ranging from leaders of NDP working groups to employees in Alzheimer’s associations. However, the perspective of the WHO remained absent because of a lack of responses from prospective interviewees. To form a complete picture, more respondents from the different organisations should be interviewed, along with respondents from more levels in society, including the patients themselves. Furthermore, the role of clinicians should be further explored, given their crucial role in diagnosing dementia. Fourth, on average, qualitative studies use more self-reporting data, which can rarely be objectively verified. Even so, our study matched quite well with the national documents examined in our multi-method analysis. To minimise that limitation, transparency was improved by using a clear methodology, conducting member checks, and discussing reflective notes during the interviews.

The role of research institutes in developing and monitoring NDPs is important for assessing the perspective of the stakeholders involved. However, representatives of such institutes were unwilling to participate in our study and cited a lack of knowledge about implementing NDPs because they focus on fundamental research on dementia, interventions, and/or medication. In Germany, although applied research is conducted, they still lack conclusive results.

Despite our worry that the experiences of respondents had been influenced by the COVID-19 pandemic, in Germany, for example, although Action Area 2 (i.e., awareness and friendliness) was overshadowed by the pandemic, especially the national dementia awareness campaign, the other action areas remained independent of the pandemic. Last, the second driver of the framework (i.e., individual, group, organisational, and systemic factors) requires interviews with more respondents at different systemic levels to clarify the particular factor and its related subfactors. Nevertheless, in our results, strong trends regarding the factors were identified.

## 5. Conclusions

Our findings demonstrate the influence of certain factors on the context and process of implementing NDPs according to different stakeholders. Such knowledge is pivotal to improving the implementation of NDPs and ensuring proper dementia care. The three countries exhibited differences in stakeholder involvement, fragmentation in and between regions, and funding for the NDPs. The study also revealed that resources, the use of change theory, and the monitoring of implementation and fragmentation are crucial factors to consider for countries when implementing NDPs. Taking those factors into account could yield better outcomes, as could the WHO’s inclusion or support of alternative factors within its framework.

As for recommendations for implementing NDPs, (local) administrators should first maintain focus on preserving the collaboration and inclusion of all stakeholders and actors involved in NDPs. At the same time, stakeholders and actors involved need to collaborate with (local) administrators as well as with each other. Second, patients with dementia, and their informal caregivers, should be included more in the design, implementation, and evaluation of NDPs in order to improve their usefulness and effectiveness. Without their participation, any NDP’s successes will not reach the people who are supposed to benefit from the policies. Third, an overarching recommendation for all parties involved in national dementia policies is to think collectively at the European, or even international, level. That process can ideally be facilitated by the WHO such that experiences about preventing and treating dementia can be shared to improve the lives of people with dementia and their informal caregiver(s), and to reduce regional inequalities under a unified approach. Finally, demand exists for empirical data and supporting research to promote and create a shared platform to increase the comparability of data and improve learning across countries—for instance, by collecting information via large-scale research, (inter)national focus groups, surveys, and data monitoring. Ultimately, follow-up studies can serve as a reality check for the WHO to make their framework and recommendations more concrete. Altogether, to make dementia care future-proof, countries need to work together in as many ways as possible.

## Figures and Tables

**Figure 1 ijerph-18-10220-f001:**
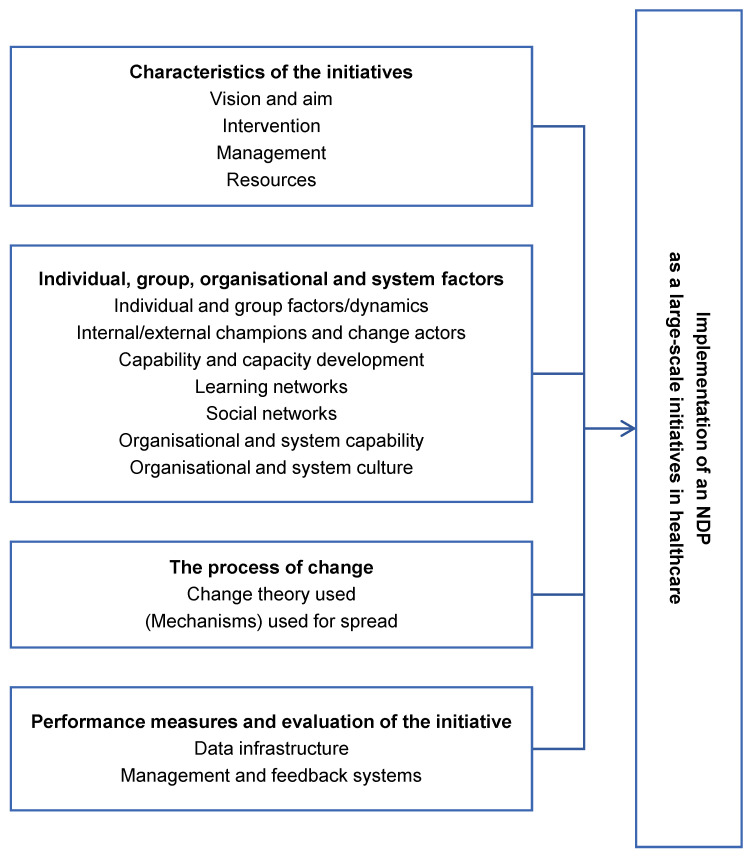
Driver diagram of large-scale change (adapted from ref. [17]).

**Table 1 ijerph-18-10220-t001:** Coded documents concerning the NDPs from the three countries.

**Denmark**	
DD1	Danish Ministry of Health (Sundhedsministeriet), “A Safe and Dignified Life with Dementia: National Action Plan on Dementia 2025”, 2017 [23].
DD2	Danish Health Authority (Sundhedsstyrelsen), “National Research Strategy on Dementia 2025”, 2018, [24]
DD3	Danish Health Authority (Sundhedsstyrelsen), “Life with Dementia–Strengthening the Quality of Effort” (Livet med demens–styrket kvalitet i indsatsen),2016 [25]
**Germany**	
DG1	Federal Ministry for Family, Seniors, Women, and Youth and Federal Ministry for Health in Germany (Bundesministerium für Familie Senioren Frauen und Jugend and Bundesministerium für Gesundheit), “National Dementia Strategy”, 2020 [26]
DG2	National Dementia Strategy Office, “National Dementia Strategy: Concept for Monitoring, ‘Network National Dementia Strategy’, and public relations” (Nationale Demenzstrategie: Konzept zu Monitoring, “Netzwerk Nationale Demenzstrategie” und Öffentlichkeitsarbeit) [27]
**Italy**	
DI1	Di Fiandra T, Canevelli M, Di Pucchio A et al. The Italian Dementia National Plan [28]
DI2	Presidency of the Council of Ministers (Presidenza del Consiglio dei Ministri), “National Dementia Plan” (Piano Nazionale Demenze), 2014 [29]
DI3	Table for monitoring the transposition and implementation of the National Dementia Plan (Tavolo per il monitoraggio del recepimento e implementazione del Piano Nazionale Demenze), “Recommendations for Governance and Clinic in the Dementia Sector” (Raccomandazioni per La Governance e La Clinica nel Settore delle Demenze), 2020 [30]
DI4	Conference of Regions and Autonomous Provinces (Conferenza delle regioni e delle province autonome), “National Guidelines for the Construction of ‘Communities Friendly to People with Dementia’” (Linee di indirizzo nazionali per la costruzione di “Comunità amiche delle persone con demenza”), 2019 [31]
DI5	Table for monitoring the transposition and implementation of the National Dementia Plan (Tavolo per il monitoraggio del recepimento e implementazione del Piano Nazionale Demenze), “National Guidelines for the Use of Information Systems to Characterise the Phenomenon of Dementia” (Linee di indirizzo Nazionali sull’uso dei Sistemi Informativi per caratterizzare il fenomeno delle demenze), 2017 [32]
DI6	Table for monitoring the transposition and implementation of the National Dementia Plan (Tavolo per il monitoraggio del recepimento e implementazione del Piano Nazionale Demenze), “National Guidelines for Diagnostic Therapeutic Assistance Pathways for Dementia” (Linee di indirizzo Nazionali sui Percorsi Diagnostico Terapeutici Assistenziali per le demenze), 2017 [33]

**Table 2 ijerph-18-10220-t002:** Respondents sorted by stakeholder group (respondent codes).

Type of Respondent	Denmark (A)	Germany (B)	Italy (C)
Ministry involved	Danish Health Authority, Department of the Elderly and Dementia (A1)	Federal Ministry for Health in Germany (B1)	Ministry of Health (C1)
Working group or expert group	Danish working group, Frederiksberg municipality (A2)	German working groups for different action fields in the Federal Ministry for Family, Seniors, Women and Youth (B2)	The Italian Dementia National Plan Working Group, Higher Institute of Health (C2)
Patients’ side	The Danish Alzheimer’s Association (A3)	German Alzheimer’s Society (B3),	Alzheimer Federation Italy (C3)
Providers’ side	Gladsaxe municipality (A4)	German Centre of Gerontology, National Dementia Strategy Office (B4)	Dementia in General Medicine (C4)

**Table 3 ijerph-18-10220-t003:** The NDPs of Denmark, Germany, and Italy alongside the GADP action areas (WHO, 2017).

GADP Action Area	Denmark (DD1)	Germany (DG1)	Italy (DI1)
1. Dementia as a public health priority?	Yes, approved NDP	Yes, approved NDP	Yes, approved NDP, with all regions involved and all accepting the NDP
2. Dementia awareness and friendliness?	Partly, budget allocated for local information campaigns and dementia-friendly initiatives but no national campaign	Yes, national campaign and dementia-friendly initiatives	Partly, promotion of a mass media dementia awareness-raising campaign and inclusion of dementia-friendly initiatives
3. Dementia risk reduction?	Yes, importance addressed, given inclusion in research and multiple national action plans to combat risks as stated in the GAPD	Yes, supported in research and addressed in the “Strategy Document of the German Alliance Against Non-Communicable Diseases for Primary Prevention”	Yes, addressed in “National Prevention Plan 2014/2018”
4. Dementia diagnosis, treatment, care, and support?	Yes, focus on equal treatment across regions	Yes, focus on improving health and long-term care services	Yes, NDP addresses the importance of timely diagnosis and the availability of services to do so
5. Support for dementia carers?	Yes, national toolbox of courses for patients and relatives	Yes, increase training for family caregivers	Yes, training programmes are available via associations and the Ministry of health
6. Information system for dementia?	Partly, a budget is provided to monitor relevant indicators for dementia but not for all initiatives	Yes, yearly questionnaire to all stakeholders in implementing a specific measure, with indicators in a later report (DG2)	No, but a round table exists for evaluating and monitoring the NDP’s implementation, and within 7 years, two surveys were issued
7. Dementia research and innovation?	Yes, recent articles are published on PubMed and area addressed in the NDP	Yes, recent articles are published on PubMed, and the area is addressed in the NDP	Yes, recent articles are published on PubMed that focus on patients’ access to care during the COVID-19 pandemic and dementia in the migrant population as part of the NDP

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
