# Peer review of "The Implementation of National Dementia Plans: A Multiple-Case Study on Denmark, Germany, and Italy"

_ijerph, 2021, doi:10.3390/ijerph181910220_

Round 1

Reviewer 1 Report

Brief summary

This study is examining and evaluating how countries developed their National Dementia Plans using a theoretical framework by Perla et al. Their primary research questions are: How do the stakeholders involve in the implementation of a national dementia plan evaluate the context of influencing factors in their country when using the WHO’s seven action areas of dementia policy? Through document abstraction and interviews, the researchers found similarities and differences in the NDP by each Country. They provided recommendations on what can be done to improve the development of NDP moving forward.

Broad comments

Overall, this is well written and an important study that contributes great information for the future development of NDPs. However, there are some notable areas where this paper could be strengthened; namely more details in the methods section.

Specific comments

Major comments

Methods:

  • Section 2.3.1. is on the document analysis. What did you do with the document once they were found? Did you extract information? Use it to create the interview questions? More clarity is needed.
  • Section 2.3.2. is on semi-structured interviews. Who were the interviewers and the interviewees? Are the interviewees the respondents from 2.3.3.? Please make this clearer.
  • What topics were discussed during these interviews? General domains would be beneficial. If possible, add a table or supplemental table with the semi-structured interview questions.
  • Line 185 states you only kept the most relevant codes. How was this determined?
  • You mentioned that your study was guided by Perla et al theory. Can you please add in the methods section what you used from the theory to help develop questions and capture data?

Discussion:

  • Are there any studies to support or that contradicted any of the statements you made throughout the discussion? For example, is there a study demonstrating an increase in dementia cases the year(s) after the initiation of Demark’s NDP? Also, is there any previous studies that analyzed any other NDP? By discussing your results and findings in the context of the literature will provide better insight to the reader on how impactful your results are.

Minor comments

Abstract:

  • Spell out the first NDP on line 10

Introduction:

  • The new sentence on line 79 is a bit confusing, rewording may be needed.

Methods:

  • On lines 132 you state “By using a constant systematic way of researching the different cases in the same manner, generalisability can be reached.” Can you please elaborate with another sentence or two how this is gained? Even though you use the same criteria to review something, there are still ecological difference on how these countries are different and generalizability may not be reached.
  • Please list all databases you search for reproducibility purposes (line 152)
  • Please give a sample searched terms used to search the databases (section2.3.1)
  • In the results you mention the seven GADP and how each country fits within these categories; however, this is analysis is not mentioned, or not clearly stated, in the methods.

Results:

  • Table 3 should be standalone. Please spell out NDP and GADP in the title.

Discussion:

  • The paragraph starting on line 428 could be combined with the next paragraph as the are related based on the stakeholder involvement.
  • In the limitations on line 453 you mention sample size, but you did not provide this in your methods or results. Please mention this earlier in the paper.

Author Response

Thank you for sharing the reviewers’ comments. We have addressed the reviewers’ concerns as follows:

  • We have expanded the Abstracts to 200 words.
  • We have included more details on the methods and explained in detail how the data was collected and analyzed as requested by reviewer 1. In addition, we have attached the interview guideline in the appendix.
  • We added a more extensive explanation on how the Perla et al. theory framework was used to derive the questions.
  • The limitation section now includes suggestions made by reviewer 2 to highlight the boundaries of our research.
  • We strengthened our recommendations on a unified approach for NDPs to make them more salient as suggested by reviewer 2.
  • Some passages were rewritten for clarity. The manuscript was further proof-read by a native speaking editor (see Confirmation of Proofreading).

We hope that our improved manuscript meets your requirements.

Response reviewer 1:

We are very thankful for the helpful suggestions. To address your comments, we have added a more detailed explanation in the Methods, including the process of the data collection, the respondents (indeed the interviewees from section 2.3.2),  and we clarified the analysis. We furthermore added a more in-depth explanation on how the Perla et al. framework was used and added the topic list for interviews in an Appendix.

We also addressed all your minor comments in terms of spelling out the acronyms and clarified some passages. Please find the relevant changes in the manuscript.

To address your comment on the discussion section “Are there any studies to support or that contradicted any of the statements you made throughout the discussion? For example, is there a study demonstrating an increase in dementia cases the year(s) after the initiation of Denmark’s NDP? Also, is there any previous studies that analysed any other NDP? By discussing your results and findings in the context of the literature will provide better insight to the reader on how impactful your results are”:

We have conducted extensive research but are not aware of any other studies that would show an increase in cases after the initiation of Denmark’s NDP. Current studies focus mostly on how aspects of NDPs can be applied to their own country. A clear evaluation of such programs is still missing or currently an on-going process (as described in the manuscript in the case of Germany). For this reason, we strongly recommend a more unified evaluation and accompanied research on NDPs to further gain empirical evidence of the usefulness of NDPs.

Thank you for your constructive feedback and hope you will find our manuscript much improved.

Reviewer 2 Report

I found this study of general interest and relatively sound in its methodological backgorund. Authors have already listed a number of limitations which temper down the possibility of generalizing the results and conclusions they draw.

I would only underline some additional aspects of particular interest (at least to this Referee):

1) the point of view of Clinicians is underestimated probably in all the three national plans, certainly in the one from Italy. General practitioneers are not the main diagnostic pillar but only a liaison step towards more specialized centres where Patients are dignosed and followed-up after diagnosis.

2) Information and continuous education activities in particular for the territorial health Opertors is also underestimated, since the field of early diagnosis is rapidly evolving.

3) In none of the three plans (and also in the WHO document) there is a specific strategy is even envisaged for identification of "prodromal to dementia"   stages in case of arrival of a disease-modifying drug approved for pre-clinical dementia stages. 

4) None of the three documents dedicates attention to the role of biomarkers and the eventual national organization for their application on a nation-wide project.

5) The importance of a unified/homogeneous approach should be strongly recommended. In the field of dementia, local/regional differences introduce a risk of inequality which should be fighted.

Author Response

Thank you for sharing the reviewers’ comments. We have addressed the reviewers’ concerns as follows:

  • We have expanded the Abstracts to 200 words.
  • We have included more details on the methods and explained in detail how the data was collected and analyzed as requested by reviewer 1. In addition, we have attached the interview guideline in the appendix.
  • We added a more extensive explanation on how the Perla et al. theory framework was used to derive the questions.
  • The limitation section now includes suggestions made by reviewer 2 to highlight the boundaries of our research.
  • We strengthened our recommendations on a unified approach for NDPs to make them more salient as suggested by reviewer 2.
  • Some passages were rewritten for clarity. The manuscript was further proof-read by a native speaking editor (see Confirmation of Proofreading).

We hope that our improved manuscript meets your requirements.

Response Reviewer 2:

Thank you for your enthusiasm for our study. We agree with your additional limitations and recommendations and have added them to the manuscript.

Your first comment about the underestimation of the point of view of Clinicians is correct, and almost fully lacking in the present NDPs. The same hold for your comments 3 and 4. We have added this to the limitations.

In comment 2 you address: “Information and continuous education activities in particular for the territorial health Operators is also underestimated, since the field of early diagnosis is rapidly evolving.”

While we agree with you, our current data does not provide clear evidence for us to derive this result. We believe this has great potential for future research together with a deeper investigation into the role of clinicians as well as patients and their caregivers.

Thank you for reviewing our manuscript and we hope you will find it much improved.

Round 2

Reviewer 1 Report

No more comments. The authors have addressed my concerns sufficiently.